# Taking the opportunity of COVID testing to screen vulnerable populations for hepatitis B, hepatitis C, syphilis, and human immunodeficiency virus in Central Brazil

Gabriel Francisco da Silva Filho[1ᵒ], Antoninho Barros Milhomem[2‡],
Bruno Vinícius Diniz e Silva[2‡], Kamila Cardoso dos Santos[1ᵒ],
Grazielle Rosa da Costa e Silva[1‡], Larissa Silva Magalhães[1‡],
Winny Éveny Alves Moura[1‡], Thaynara Lorrane Silva Martins[1‡],
Wanessa de Oliveira Gonçalves[1‡], Marcia Alves Dias Matos[2‡],
Roxana Isabel Cardozo Gonzales[1‡], Leonora Rezende Pacheco[1‡],
Regina Maria Bringel Martins[2‡], Karlla Antonieta Amorim Caetano[1ᵒ],
Megmar Aparecida dos Santos Carneiro[2ᵒ], Sheila Araújo Teles[1ᵒ]*

**1** Faculty of Nursing, Federal University of Goiás, Goiânia, Goiás, Brazil, **2** Institute of Tropical Medicine and Public Health, Federal University of Goiás, Goiânia, Goiás, Brazil

ᵒ These authors contributed equally to this work.
‡ ABM,BVS,GRDCS, LSM, WEAM, TLSM, WDOG and MADM, RICG, LRP,RMBM, also contributed equally to this work.
* sheila.fen@gmail.com

## Abstract

Vulnerable populations were disproportionally affected by the coronavirus disease 2019 (COVID-19) pandemic and its peak periods disrupted progress toward the control and prevention of sexually transmitted infections (STIs). This study aimed to investigate the prevalence and factors associated with hepatitis B virus (HBV), hepatitis C virus (HCV), syphilis, and human immunodeficiency virus (HIV) among socio-economically vulnerable populations during the COVID-2019 pandemic in Goiania, Central Brazil. A total of 627 individuals, including lesbian, gay, bisexual, and transgender people, homeless individuals, waste recyclable collectors, and immigrants/refugees, were tested for these infections. Multiple Poisson regression models were used to analyze the factors associated with each outcome. HBV exposure markers were found in 16.1% of participants, antibodies against HCV (anti-HCV) in 1.9%, syphilis in 17.2%, and antibodies against HIV (anti-HIV) in 6.1%. Thirty-two individuals exhibited serological evidence of active syphilis, whereas HBV deoxyribonucleic acid and HCV ribonucleic acid were detected in six and two individuals, respectively. Increased age was statistically associated with HCV, HBV, and syphilis; identifying as LGBT and reporting a history of STIs was associated with syphilis and HIV. Illicit drug use was associated with HCV and HIV, whereas being an immigrant/refugee or engaging in transactional sex was linked to HBV. Homelessness and reporting a higher number of sexual partners in the previous month were associated

**Data availability statement:** All relevant data are within the manuscript and its Supporting Information files.

**Funding:** Fundação de Amparo à Pesquisa do Estado de Goiás - FAPEG, CHAMADA FAPEG No 05/2020.

**Competing interests:** NO authors have competing interests.

with syphilis. High prevalence rates of these infections were identified, with many participants showing evidence of active infections, increasing the risk of transmission. Specific risk behaviors were associated with each infection, emphasizing the need to tailor prevention strategies to address these behaviors effectively.

## Introduction

Human immunodeficiency virus (HIV), viral hepatitis B and C (HBV and HBC), and sexually transmitted infections (STIs) represent significant global public health concerns and are key targets of the Global Health Sector Strategies 2022–2030. An estimated 3.5 million new cases of HIV, HBV, and HCV are reported annually, whereas one million STIs are diagnosed each day. These infections cause high morbidity and mortality, collectively responsible for 2.5 million deaths and 1.2 million cases each year [1].

HIV causes acquired immunodeficiency syndrome (AIDS) [2]; however, early treatment can prevent AIDS progression [3]. HBV and HCV are significant causes of cirrhosis and hepatocellular carcinoma [4]. A safe and effective HBV vaccine has been available since the 1980s [5], and direct-acting antiviral (DAA) treatments now offer a cure for hepatitis C [6]. Although syphilis can be treated with low-cost antibiotics, the infection remains a cause of neurological and cardiovascular complications in individuals infected with *T. pallidum* [7]. These pathogens share common modes of transmission and prevention measures [1].

Available vaccines, curative treatments, and preventive measures create opportunities to eliminate HIV, viral hepatitis, and syphilis, aligning with the World Health Organization's (WHO) goal to reduce these infections as public health threats by 2030. However, considerable efforts should be made to diagnose infected people and link them to health services for treatment and prevention [8].

In 2023, an estimated 86% of individuals living with HIV knew their serostatus, and 89% of all people living with HIV were receiving antiretroviral therapy. Conversely, by the end of 2022, only 13.4% of the people living with HBV and 36.4% of those with HCV were aware of their diagnoses, with 2.6% and 20% of diagnosed individuals receiving treatment, respectively [8]. Notably, most individuals living with these infections experience social stigma, inequities, and limited access to health services, which hinders early diagnosis and treatment [8].

The coronavirus disease 2019 (COVID-19) pandemic disproportionally affected economically vulnerable populations, and its peak periods disrupted progress in controlling and preventing STIs. Therefore, WHO recommended taking advantage of COVID-19 testing initiatives to screen vulnerable people for HIV, viral hepatitis, and STIs [8].

In Brazil, during the first and second waves of the COVID-19 pandemic, our research utilized COVID-19 testing initiatives to screen vulnerable people for HIV, viral hepatitis, and STIs, as recommended by WHO (WHO, 2021b). Therefore, this study shows the prevalence and factors associated with HIV, syphilis, HBV, and HCV in vulnerable populations during the first and second waves of COVID-19 in Goiânia, Central Brazil.

## Materials and methods

### Study design, location, population, and sample

This cross-sectional study was conducted in Goiânia, located in Central Brazil, with a population of 1,437,237 inhabitants. The study focused on socially and economically vulnerable groups, including lesbian, gay, bisexual, and transgender (LGBT) people, immigrants/refugees, homeless people, and recyclable waste collectors.

Individuals aged 14 years or older who were assisted by non-governmental organizations (NGOs) catering to the needs of LGBT people, immigrants and refugees, recyclable waste collectors, and homeless individuals were included in the study. Exclusion criteria comprised individuals who appeared to be under the influence of alcohol or illicit drugs during the recruitment process.

The prevalence of HCV previously reported among recyclable waste collectors was considered for sample calculation [8]. Therefore, the required sample size for the study, considering a 95% significance level, a precision of 1.0%, and an anti-HCV prevalence of 1.6% [9], was 605 individuals.

### Data collection

In Brazil, at the beginning of the pandemic, COVID-19 testing by public health services was limited for the general population, particularly vulnerable people. Therefore, from August 1st, 2020, to April 28th, 2021, the Federal University of Goiás offered free COVID-19 testing for socially and economically vulnerable individuals assisted by NGOs. We took advantage of this window of opportunity and invited them to participate in the study on HIV, syphilis, and viral hepatitis B and C. Data collection occurred on the University dependencies following biosafety practices. All eligible individuals were invited to participate in the study and provided information regarding the study objectives and procedures. Those who agreed to participate read and signed a written informed consent (IC). Their parents or legal guardians authorized the participation of individuals aged less than 18 years, and a particular written IC was read and signed. In the case of illiteracy, the IC was read to the participant and witness, and the signature was performed by dactyloscopy.

Face-to-face interviews were conducted using a previously validated instrument on sociodemographic and risk factors for STIs. For immigrants and refugees who did not speak or understand Portuguese, the IC and instrument were translated into French, Haitian Creole, and Spanish, and interviews were conducted by duly trained translators in Creole, French, or Spanish.

After completing the interviews, blood samples (10 mL) were collected from participants to detect the markers of HIV, HBV, HCV, and syphilis. Initially, rapid diagnostic tests were employed, following the manufacturer's instruction, to detect hepatitis B surface antigen (HBsAg) using the HBsAg Bioclin test (Quibasa Química Básica, Belo Horizonte, MG), antibodies to HCV (anti-HCV) using the HCV ab ECO test (Biosensor, ECO Diagnóstica, Corinto, MG), and syphilis (ABON™ Syphilis Ultra Rapid Test, Abbott). For anti-HIV types 1 and 2, initial screening was performed using the ABON™ HIV 1/2/O Tri-Line Human Immunodeficiency Virus Rapid Test (Abbott), with reactive samples confirmed using the TR DPP HIV 1/2 (Bio-Manguinhos).

All samples underwent additional testing via an enzyme-linked immunosorbent assay (ELISA) for HBsAg (BIOELISA HBsAg, Bioclin, Quibasa Química Básica, Belo Horizonte, MG), total antibodies to hepatitis B core antigen (anti-HBc total) (Elisa anti-HBc, Wieber Lab., Rosario, Argentina), antibodies to hepatitis B surface antigen (anti-HBs) (BIOELISA Anti-HBs, Bioclin, Quibasa Química Básica, Belo Horizonte, MG), and anti-HCV (BIOELISA HCV, Bioclin, Quibasa Química Básica, Belo Horizonte, MG). Samples reactive for syphilis were further analyzed using the venereal disease research laboratory (VDRL) test (RPR Brás, Laborclin, Pinhais, Paraná). Active syphilis was diagnosed in individuals with VDRL titers ≥ 1/8 in the non-treponemal test.

Samples positive for HBsAg and/or anti-HBc total were subjected to deoxyribonucleic acid (DNA) extraction using the QIAamp® Blood Mini Kit (Qiagen, Hidden, Germany). The pre-S/S region of the HBV genome was amplified through a

semi-nested polymerase chain reaction (PCR) assay with a detection limit of three copies of HBV DNA per reaction, as previously described [10,11].

HCV ribonucleic acid (RNA) was extracted from samples positive for anti-HCV (QIAamp® Viral RNA Mini Kit; Qiagen, Hilden, Germany), and detected using reverse transcription (RT) nested PCR with primers targeting the conserved area of the 5' non-coding region of the HCV genome, as outlined in previous studies [12].

### Data analysis

The data were collected, entered into EpiData version 3.1, and exported to STATA version 17 SE. Some individuals belong to more than one vulnerable group. Then, for the variable "population," the overlap in the total number of people in each group was considered. Further, for HBV analysis purposes, individuals who presented previous hepatitis B vaccine serological profiles (anti-HBs only positive) were considered in the denominator.

Descriptive statistics, including absolute and relative frequencies and measures of central tendency, were used. Parametric and non-parametric tests were used to analyze the differences between medians and proportions, as appropriate. The Poisson multiple regression model included variables with $p < 0.20$. Statistical significance was set at $p < 0.05$. Goodness-of-fit tests were used to evaluate the quality of the models.

The Research Ethics Committee of *Universidade Federal de Goiás* approved this study (protocol number 4.249.851).

## Results

### Characteristics of participants

Six hundred twenty-two individuals were eligible and agreed to blood collection. The participants included 279 recyclable waste collectors, 153 immigrants/refugees, 117 homeless people, and 78 LGBT people (Table 1). Some participants were categorized into more than one group. The majority were male (53%), mixed race (52.8%), or single (58.4%). The median age was 34 years (interquartile range [IQR]: 20), the median monthly income was 1,200 BRL (IQR: 976), and the median years of education were 10 (IQR: 6). Additionally, 14.6% of the participants tested positive for COVID-19 (Table 1).

Of the total participants, 31 (4.9%) reported a history of physical violence, and two (0.3%) reported coercive sex. A history of STIs was found in 19.4% of participants, and 10.3% reported engaging in transactional sex. Additionally, 39.4% of participants reported engaging in anal sex, and 59.9% did not use condoms during their last sexual intercourse. Nearly one-third of the participants (31.9%) reported a history of illicit drug use. Daily alcohol consumption was reported by 4.6% of participants, and 15.9% had experienced arrest. Moreover, 4.5% of participants were aware of their HIV diagnosis.

The characteristics per group are shown in the S1 Table. The median age ranged from 30 years old (Immigrant/refugee and LGBT people) to 37 for waste recycler pickers, respectively, while the monthly income ranged from 1,150 R$ for the homeless to 2,000 R$ for LGBT people. Time of scholarly was lower among homeless people and recycling waste collectors (median: 9 years) and higher among immigrants/refugees and LGBT people (median: 12 years). Transactional sex was more frequent among LGBT people (28.9%), followed by homeless people (15.9%) and waste recycler pickers (7.5%). Only one immigrant/refugee reported this behavior. Non-condom use was frequent in all groups, with a higher proportion among immigrants/refugees (74.8%). STI report and anal sex varied from 40.4% and 79.8% among LGBT people to 5.6% and 18.1% among immigrants/refugees, respectively. Daily alcohol use ranged from 10% among the homeless to 1.2% among immigrants/refugees, but illicit drug use varied from 57.6% among LGBT people to 7.4% among immigrants/refugees. Previous arrests were more frequent among homeless people (32.6%) and waste recycler picker (14.6%).

### Prevalence of HBV, HCV, syphilis, and HIV markers

HBV exposure markers (HBsAg and anti-HBc) were found in 101 (16.1%) individuals, anti-HCV in 1.9%, anti-HIV in 6.1%, and syphilis in 17.2%. Among 108 individuals who tested positive for the treponemal test, 32 (29.63%) also tested positive for the VDRL test (≥ 1/8) (Table 2).

**Table 1. Characteristics of the 627 vulnerable people in Goiânia, Central Brazil.**

| Category variables | n | % |
|---|---|---|
| **Vulnerable population*** | | |
| Waste recycle pickers | 279 | 44.5 |
| Immigrants/refugees | 153 | 24.4 |
| Homeless people | 117 | 18.7 |
| LGBT | 78 | 12.4 |
| **Gender** | | |
| Male | 332 | 53.0 |
| Female | 295 | 47.0 |
| **Race/ethnicity** | | |
| White | 111 | 17.7 |
| Black | 146 | 23.3 |
| Mixed | 331 | 52.8 |
| Asian/indigenous | 39 | 6.2 |
| **Marital status** | | |
| Single | 362 | 58.4 |
| Married | 212 | 34.2 |
| Widow/divorced/separated | 46 | 7.4 |
| No information; 7 | | |
| **COVID-19 #** | | |
| No | 517 | 83.0 |
| Yes | 91 | 14.6 |
| Indeterminate | 15 | 2.4 |
| **Aware of HIV diagnosis** | | |
| No | 599 | 95.5 |
| Yes | 28 | 4.5 |
| **Report of physical violence** | | |
| No | 596 | 95.1 |
| Yes | 31 | 4.9 |
| **Report of coercive sex** | | |
| No | 625 | 99.7 |
| Yes | 2 | 0.3 |
| **Anal sex** | | |
| No | 366 | 60.6 |
| Yes | 238 | 39.4 |
| No information; 23 | | |
| **Condom use in the last sexual intercourse** | | |
| Yes | 238 | 40.1 |
| No | 355 | 59.9 |
| No information; 34 | | |
| **STI[b] report** | | |
| No | 495 | 80.6 |
| Yes | 119 | 19.4 |
| No information; 13 | | |
| **Transactional sex** | | |
| No | 547 | 89.7 |
| Yes | 63 | 10.3 |

*(Continued)*

**Table 1.** (Continued)

| Category variables | n | % |
|---|---|---|
| No information; 17 | | |
| **Illicit drug use** | | |
| No | 424 | 68.1 |
| Yes | 199 | 31.9 |
| No information; 4 | | |
| **Daily alcohol consumption** | | |
| No | 598 | 95.4 |
| Yes | 29 | 4.6 |
| **Previous arrest** | | |
| No | 517 | 84.1 |
| Yes | 98 | 15.9 |
| No information; 12 | | |
| **Continuous variables** | **Median** | **IQR**[a] |
| Age | 34 | 20 |
| Monthly income (R$[c]) | 1,200 | 976 |
| Years of education | 10 | 6.0 |
| Number of sexual partners (last month) | 1 | 0 |

Legend:

*Nine were recyclable waste collectors and immigrants/refugees; four were recyclable waste collectors and homeless people; nine were recyclable waste collectors and LGBT people; 12 were homeless and LGBT people; and three were LGBT and immigrants/refugees.

# Four individuals denied nasopharyngeal swab.

[a] IQR: interquartile range.

[b] STI: sexually transmitted infection.

[c] R$ 5,64 was equivalent to US$1.

## Factors associated with HBV, HCV, syphilis, and HIV

In the bivariate analysis, considering the HCV outcome, the variables, including, homelessness, age, schooling, monthly income, anal sex, condom use, STI report, illicit drug use, previous arrest, and awareness of HIV diagnosis showed a $p$-value < 0.20 (S2 Table) and were included in a multiple regression model. The final model showed that for each year of age, anti-HCV prevalence increased by 8.5% (adjusted prevalence ratio [APR]: 1.085; 95% CI: 1.041–1.130). Individuals who used illicit drugs presented an APR of 5.858 (95% CI: 1.212–28.312) compared to non-drug users. Moreover, individuals who were aware of their HIV diagnosis showed an APR of 3.673 (95% CI: 1.391–9.695) than their counterparts (Table 3).

HBV exposure was significantly associated with being waste recycling pickers, immigrants/refugees, age, STI reports, and awareness of one's own HIV diagnosis ($p$ < 0.05) (S2 Table). These variables, along with being LGBT, number of sexual partners in the last month, non-condom use, transactional sex, and drug use were included in a multiple Poisson regression model, and being immigrants/refugees (APR: 2.582; 95% CI: 1.179–3.724), aging (APR: 1.023; 95% CI: 1.011–1.035); awareness of own HIV diagnosis (APR: 2.204; 95% CI: 1.452–3.344) and transactional sex (APR: 1.907; 95% CI: 1.291–2.816) remained independently associated with HBV markers (Table 3).

Initially, syphilis was associated with all vulnerable groups studied plus age, physical violence, number of sexual partners in the last month, anal sex, non-condom use, STI report, transactional sex, illicit drug use, daily alcohol consumption, previous arrest, and awareness of HIV diagnosis (S3 Table). Multiple regression analysis included these variables

**Table 2. Prevalence of HBV, HCV, syphilis, and HIV serological markers among vulnerable people in Goiânia, Central Brazil.**

| Markers | Pos./total | (%) | 95% CIª |
|---|---|---|---|
| **HBV** | | | |
| **HBsAg+** | 2/627 | 0.32 | 0.09–1.16 |
| **HBsAg+Anti-HBc+** | 5/627 | 0.80 | 0.34–1.85 |
| **Anti-HBc only+** | 20/627 | 3.19 | 2.07–4.86 |
| **Anti-HBc+/anti-HBs+** | 74/627 | 11.96 | 9.65–14.74 |
| **Any exposure marker** | 101/627 | 16.11 | 13.44–19.19 |
| **Anti-HBs (immunized)+** | 250/627 | 39.87 | 36.11–43.76 |
| **HCV** | | | |
| **Anti-HCV+** | 12/627 | 1.91 | 1.10–3.32 |
| **HIV** | | | |
| **Anti-HIV+** | 38/627 | 6.1 | 4.45–8.21 |
| **Syphilis** | | | |
| **Anti-*T. Pallidum*+** | 108/627 | 17.2 | 14.47–20.38 |
| **VDRL ≥ 1:8+** | 32/108 | 29.63 | 21.84–38.82 |

Legend:

ª Confidence Interval.

HBV DNA was detected in six of 101 anti-HBc positive samples: three anti-HBc positive/HBsAg positive and three anti-HBc positive/HBsAg negative (occult HBV infection). HCV RNA was detected in two out of 12 anti-HCV positive samples.

and the number of sexual partners in the last month ($p = 0.150$). The variables associated with syphilis in the final model were: being homeless people (RPA: 1.866; 95% CI: 1.304–2.670), being LGBT (RPA: 1.919; 95% CI: 1.349–2.731), aging (RPA:1.012; 95% CI: 1.001–1.025), increased number of sexual partners in the last month (RPA: 1.001; 95% CI: 1.000–1.002), and report of STI 3.371 (95% CI: 2.280–4.985), whereas condom use in the previous sexual intercourse was a protective factor (RPA: 0.683; 95% CI: 0.482–0.967) (Table 3).

All vulnerable groups were also associated with HIV in bivariate analysis, besides anal sex, non-condom use, STI report, transactional sex, illicit drug use, and previous arrest (S4-S5 Tables). The final model showed that LGBT people (RPA: 2.209; 95% CI: 1.072–4.553), STI report (RPA: 12.076; 95% CI: 4.577–31.859), and illicit drug use (RPA: 1.889; 95% CI: 1.041–3.452) remained associated with HIV (Table 3).

## Discussion

The COVID-19 pandemic was the worst health crisis of the 21st century. In addition to the direct effects of the novel coronavirus, which has caused elevated morbidity and mortality globally, several health programs have been compromised. Therefore, optimized efforts are required to overcome this health sector crisis [13]. We took advantage of COVID-19 screening and tested vulnerable individuals for HBV, HCV, syphilis, and HIV. Additionally, active infections were identified and referred to by public health services.

In this study, HIV prevalence among the vulnerable groups reached 6.1%, a rate 15 times higher than the prevalence in the Brazilian general population (0.4%) [14]. Syphilis prevalence (17.2%) significantly exceeded that of Brazilian sentinel populations, such as male conscripts (1.09%) [15] and pregnant women (1.02%) [16]. The anti-HCV prevalence was 1.91%, almost four times greater than the general population estimate of 0.53% [17]. HBV exposure markers were detected in 16.1% of participants, slightly higher than the national prevalence of 11.6%, likely reflecting the protective effect of the hepatitis B vaccine, which has been universally available in Brazil [18]. These findings underscore the disproportionate burden of STIs among vulnerable people and the necessity of creating opportunities for regular screening to meet the global goal of eliminating these infections by 2030. Moreover, the significant proportion of individuals with active

**Table 3. Variables associated with HBV, HCV, Syphilis, and HIV among vulnerable people in Goiânia, Brazil Central.**

| HCV[a] | p-value | APR (95% CI) |
|---|---|---|
| Homeless | 0.090 | 2.565 (0.862–7.632) |
| Age | < 0.001 | 1.085 (1.041–1.130) |
| Illicit drug use | < 0.028 | 5.858 (1.212–28.312) |
| Aware of HIV diagnosis | < 0.009 | 3.673 (1.391–9.695) |
| **HBV[b]** | **p-value** | **APR (95% CI)\*** |
| Immigrants/refugees | < 0.001 | 2.582 (1.791–3.724) |
| Age | < 0.001 | 1.023 (1.011–1.035) |
| Aware of HIV diagnosis | < 0.001 | 2.204 (1.452–3.344) |
| Transactional sex | 0.001 | 1.907 (1.291–2.816) |
| **Syphilis[c]** | **p-value** | **APR (95% CI)** |
| Homeless | 0.001 | 1.866 (1.304–2.670) |
| LGBT | < 0.001 | 1.919 (1.349–2.731) |
| Age | 0.043 | 1.012 (1.001–1.025) |
| Number of sexual partners in the last month | 0.023 | 1.001 (1.000–1.002) |
| STI report | < 0.001 | 3.371 (2.280–4.985) |
| Condom use in the last sexual intercourse | 0.032 | 0.683 (0.482–0.967) |
| **HIV[d]** | **p-value** | **APR (95% CI)** |
| Waste recycle pickers | 0.057 | 0.201 (0.038–1.050) |
| Homeless people | 0.274 | 1.390 (0.770–2.508) |
| LGBT | 0.032 | 2.209 (1.072–4.553) |
| STI report | < 0.001 | 12.076 (4.577–31.859) |
| Illicit drug use | 0.036 | 1.889 (1.041–3.452) |
| Condom use in the last sexual intercourse | 0.055 | 0.535 (0.283–1.041) |

Legend:

\*APR (95% CI); adjusted prevalence ratio; 95% confidence interval.

[a] Adjusted by homeless, age, years of schooling, monthly income, aware of HIV diagnosis, condom use, anal sex, STI, previous arrest and illicit drug use; [b] Adjusted by immigrants/refugees, waste recycler picker, LGBT, age, STI, aware of HIV diagnosis, condom use, transactional sex and illicit drug use; [c] Adjusted by homeless, LGBT, immigrants/refugees, waste recycler pickers, age, anal sex, physical violence, condom use, illicit drug use, number of sexual partners in the last month, aware of HIV diagnosis, previous arrest, STI, transactional sex and alcohol daily consumption; [d] Adjusted by homeless, LGBT, waste recycle pickers, immigrants/refugees, gender, physical violence, illicit drug use, previous arrest, STI, anal sex, condom use, and transactional sex.

infections and high-risk sexual behaviors emphasizes their potential role in further transmission, reinforcing the need for targeted public health interventions.

Statistical analysis revealed common variables associated with the outcomes. The association between aging and HCV, HBV, and syphilis reflects the cumulative opportunities for exposure to these agents throughout life. Being aware of an HIV diagnosis was associated with HCV and HBV infections. Viral hepatitis B and C and HIV coinfections are common. Two systematic reviews and meta-analyses estimated the global prevalence of HBV and HCV among people living with HIV. The prevalence of HCV/HIV ranges from 2.4% in the general population to 82.4% among individuals who inject drugs [19]. For HBV, HBsAg prevalence ranges from 6.1% in heterosexual/pregnant and men who have sex with men to 11.8% in people who inject drugs [20]. Individuals co-infected with HIV and viral hepatitis B and/or C are at a higher risk of accelerated progression of liver diseases and HCC [21].

LGBT people were associated with syphilis and HIV, highlighting the importance of driving preventive measures for this key population [1]. STI report was also associated with syphilis and HIV, representing a surrogate marker of unsafe sex and a sensitive question for STI screening.

Psychoactive drugs increase sexual desire, arousal, and risk-taking behavior [22]. Therefore, a high frequency of HIV and HCV has been observed among illicit drug users [23,24]. A strong association was detected between drug use and HCV and HIV. In countries where HCV screening in blood banks includes nucleic acid tests (NAT) and safety care procedures, drug injection use represents the major risk factor for HCV [25,26]. Therefore, offering HCV testing for injection drug users and linking them to health services for DAA treatment is imperative to eliminate HCV as a global public health problem [1]. In Brazil, these medicines are free of charge for all populations [27], and the challenge is to reach people who live mostly outside the reach of public health services.

Brazil has become a destination for immigrants and refugees in recent decades, mainly from Latin American countries [28]. This study identified a 2.6-fold higher APR for HBV exposure among immigrants and refugees compared to other vulnerable groups, highlighting the importance of hepatitis B vaccination and testing for these emergent population groups. HBV is efficiently transmitted sexually in countries with low endemicity, making sexual practices a significant focus for preventive measures [29].

An increased number of sexual partners in the previous month was associated with syphilis infection. This, combined with low condom use during sexual activities, represents the cornerstone of STI dissemination. Surprisingly, not using a condom during the last sexual intercourse was a protective factor against syphilis. However, this result should be considered cautiously, and reverse causality should not be ruled out. Individuals with syphilis may be more conscious of safe sex.

Homeless people were significantly associated with syphilis, consistent with previous studies showing a high prevalence in this population [30,31]. A previous investigation in Goiânia reported a lifetime syphilis prevalence of 22%, confirming their elevated risk for this infection [32].

This study had limitations. Firstly, the cross-sectional design limited the ability to establish causality between risk factors and infections. Secondly, the use of a convenience sample may not have fully represented the broader population of vulnerable individuals. Despite these limitations, the findings were consistent with those reported in the existing literature.

This study successfully tested vulnerable populations for STIs during the COVID-19 pandemic. It revealed a high prevalence of the investigated infections and identified multiple risk factors associated with their transmission. All individuals testing positive were referred to public health services specializing in STIs. Overcoming barriers in the health sector during crises remains crucial to effectively addressing the healthcare needs of at-risk populations.

## Supporting information

**S1 Table. Characteristics of vulnerable people, according to the group. in Goiânia, Central Brazil.**
(DOCX)

**S2 Table. Bivariate analysis of potential variables associated with anti-HCV among vulnerable people in Goiânia, Central Brazil.**
(DOCX)

**S3 Table. Bivariate analysis of potential variables associated with HBV among vulnerable people in Goiânia, Central Brazil.**
(DOCX)

**S4 Table. Bivariate analysis of potential variables associated with syphilis among vulnerable people in Goiânia, Central Brazil.**
(DOCX)

 

**S5 Table. Bivariate analysis of potential variables associated with HIV among vulnerable people in Goiânia, Central Brazil.**
(DOCX)

## Acknowledgments

We sincerely thank all collaborators whose contributions were vital to this study. We also appreciate the participants' generous sharing of their time and experiences, which made this research possible. We are also thankful to the institutions and facilities that kindly provided their spaces and resources, enabling the execution of experiments and sample collection. Your support and commitment have been invaluable to the success of this work.

## Author contributions

**Conceptualization:** Karlla Antonieta Amorim Caetano, Megmar Aparecida dos Santos Carneiro, Sheila Araújo Teles.

**Data curation:** Gabriel Francisco da Silva Filho, Antoninho Barros Milhomem, Bruno Vinícius Diniz e Silva, Kamila Cardoso dos Santos, Grazielle Rosa da Costa e Silva, Larissa Silva Magalhães, Winny Éveny Alves Moura, Thaynara Lorrane Silva Martins, Karlla Antonieta Amorim Caetano, Megmar Aparecida dos Santos Carneiro, Sheila Araújo Teles.

**Formal analysis:** Bruno Vinícius Diniz e Silva, Kamila Cardoso dos Santos, Larissa Silva Magalhães, Karlla Antonieta Amorim Caetano, Megmar Aparecida dos Santos Carneiro, Sheila Araújo Teles, Marcia Alves Dias Matos.

**Funding acquisition:** Sheila Araújo Teles.

**Investigation:** Gabriel Francisco da Silva Filho, Grazielle Rosa da Costa e Silva, Winny Éveny Alves Moura, Thaynara Lorrane Silva Martins, Wanessa de Oliveira Gonçalves, Roxana Isabel Cardozo Gonzales, Karlla Antonieta Amorim Caetano, Megmar Aparecida dos Santos Carneiro, Sheila Araújo Teles.

**Methodology:** Gabriel Francisco da Silva Filho, Kamila Cardoso dos Santos, Karlla Antonieta Amorim Caetano, Megmar Aparecida dos Santos Carneiro, Sheila Araújo Teles, Marcia Alves Dias Matos.

**Project administration:** Sheila Araújo Teles.

**Resources:** Sheila Araújo Teles.

**Software:** Wanessa de Oliveira Gonçalves, Sheila Araújo Teles.

**Supervision:** Karlla Antonieta Amorim Caetano, Sheila Araújo Teles.

**Validation:** Antoninho Barros Milhomem, Karlla Antonieta Amorim Caetano, Megmar Aparecida dos Santos Carneiro, Sheila Araújo Teles.

**Visualization:** Karlla Antonieta Amorim Caetano, Sheila Araújo Teles.

**Writing – original draft:** Gabriel Francisco da Silva Filho, Antoninho Barros Milhomem, Bruno Vinícius Diniz e Silva, Kamila Cardoso dos Santos, Grazielle Rosa da Costa e Silva, Larissa Silva Magalhães, Winny Éveny Alves Moura, Wanessa de Oliveira Gonçalves, Regina Maria Bringel Martins, Sheila Araújo Teles.

**Writing – review & editing:** Gabriel Francisco da Silva Filho, Antoninho Barros Milhomem, Bruno Vinícius Diniz e Silva, Roxana Isabel Cardozo Gonzales, Leonora Rezende Pacheco, Regina Maria Bringel Martins, Karlla Antonieta Amorim Caetano, Megmar Aparecida dos Santos Carneiro, Sheila Araújo Teles.

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
