## [Decision Letter · Decision Letter 0]

PONE-D-24-55277Screening vulnerable populations for hepatitis B, hepatitis C, syphilis, and human immunodeficiency virus during the coronavirus disease 2019 pandemicPLOS ONE

Dear Dr. Teles,

Thank you for submitting your manuscript to PLOS ONE. After careful consideration, we feel that it has merit but does not fully meet PLOS ONE’s publication criteria as it currently stands. Therefore, we invite you to submit a revised version of the manuscript that addresses the points raised during the review process.

The 3 reviewers found your article of interest but questioned the relationship with the COVID-19 epidemic except the opportunity to obtained samples from the vunerable population. A few more word about the expected percentage of this population that were involved in the covid survey will help to improve the significance of your data. Please also take care about the few improvment of the presentation (position of table etc...).

We look forward to receiving your revised manuscript.

Kind regards,

Pierre Roques, Ph.D.

Academic Editor

PLOS ONE

Journal Requirements:

2. Please describe in your methods section how capacity to provide consent was determined for the participants in this study. Please also state whether your ethics committee or IRB approved this consent procedure. If you did not assess capacity to consent please briefly outline why this was not necessary in this case.

Fundação de Amparo à Pesquisa do Estado de Goiás - FAPEG,  CHAMADA FAPEG No 05/2020 

Additional Editor Comments (if provided):

Reviewers' comments:

Reviewer's Responses to Questions

**Comments to the Author**

1. Is the manuscript technically sound, and do the data support the conclusions?

Reviewer #1: Yes

Reviewer #2: Yes

Reviewer #3: Yes

2. Has the statistical analysis been performed appropriately and rigorously? 

Reviewer #1: Yes

Reviewer #2: Yes

Reviewer #3: Yes

3. Have the authors made all data underlying the findings in their manuscript fully available?

Reviewer #1: Yes

Reviewer #2: Yes

Reviewer #3: Yes

4. Is the manuscript presented in an intelligible fashion and written in standard English?

Reviewer #1: Yes

Reviewer #2: Yes

Reviewer #3: Yes

5. Review Comments to the Author

Reviewer #1: The study is interesting but i have some reservations and suggestions that would potebtially improve the manuscript.

Overall much is said about the covid 19 context but in fact, although the study used the covid 19 context to do the study i think it has nothing to do with covid. Unless the authors hypothesize that the prevalences and risk factors are linked to covid it should be explicit that the study just piggybacked the covid intervention...

If the authors do believe that covid is a central part of their hypothesis they should be explicit and state how the design addresses that hypothesis.

Why were persons using alcohol or drugs excluded? One could argue that they are a vulnerable group.

The population studied seems like a composite group that is "vulnerable" but quite heterogenous

Waste collectors, homeless and lgbt are very different worlds and sexual behaviors. There seems to be an opportunistic aspect to this choice, to have a bigger sample size

For the models the risk is not necessarily linearly associated with age so i suggest using age categories to compute APR.

In the results the text repeats values and 95%CI that are already in the table. I think it is a bit redundant.

Since we are mostly talking about sexual risk there should be a table with the sexual risks per subpopulation (what proportion of waste collectors used condoms, engaged in anal sex, translational sex...etc for refugees and migrants lgbt....)

Reviewer #2: Dear authors, I have read the article entitled “Screening vulnerable populations for hepatitis B, hepatitis C, syphilis, and human immunodeficiency virus during the coronavirus disease 2019 pandemic”. I have some minor concerns:

For better understanding, the description of the data in each table should be given before the tables themselves appear (Lines: 172-178; lines: 212-237).

In addition, as each table corresponds to a part of the results, each part must have a title.

Lines 183-186: These data are not shown in Table 2.

Reviewer #3: 1. Line 1 to 3: The authors could specify the study area in the title.

2. Line 23: The 273-word abstract clearly highlights the important results of this study.

However, it would be interesting to highlight the context and objectives of the study at the beginning of the summary.

3. Harmonize this reference with the previous ones which are in the numbered style.

4. Line 144 to 147: Which extraction kit was used for HCV RNA extraction?

5. Line 154:Make the P lowercase. Harmonize with the previous one.

6. Line 167 (table 1): The sum of the percentages exceeds 100; there are certainly groups that overlap; could you separate them?

7. Conclusion: The manuscript does not clearly highlight the correlation between COVID-19 and other infections. I therefore suggest that we remove the term from the title.

I suggest to the editors that they accept the manuscript with minor revisions.

6. PLOS authors have the option to publish the peer review history of their article (what does this mean? ). If published, this will include your full peer review and any attached files.

**Do you want your identity to be public for this peer review?** For information about this choice, including consent withdrawal, please see our Privacy Policy .

Reviewer #1: No

Reviewer #2: No

Reviewer #3: **Yes: ** Jean Bienvenue OUOBA

---

## [Author Response · Author response to Decision Letter 1]

30 Apr 2025

Dear Editor,

Thank you for your careful consideration. We appreciate all the suggestions from your reviewers about our manuscript "Screening vulnerable populations for hepatitis B, hepatitis C, syphilis, and human immunodeficiency virus during the coronavirus disease 2019 pandemic (PONE-D-24-55277)". We agree with the reviewer’s comments and carefully attempted to address their questions and comments.

Reviewer #1: The study is interesting, but I have some reservations and suggestions that would potentially improve the manuscript.

Q1a Overall much is said about the covid 19 context but in fact, although the study used the covid 19 context to do the study I think it has nothing to do with covid. Unless the authors hypothesize that the prevalences and risk factors are linked to covid it should be explicit that the study just piggybacked the covid intervention...If the authors do believe that covid is a central part of their hypothesis they should be explicit and state how the design addresses that hypothesis.

R1a: We appreciate the reviewer's comment and understand the need to clarify the role of COVID-19 in our study. Our research does not hypothesize that the prevalence and risk factors of HIV, viral hepatitis, and STIs are directly linked to the COVID-19 pandemic. Instead, we leveraged the COVID-19 testing initiatives as an opportunity to screen vulnerable populations for these infections, as recommended by the WHO. To address this concern, we have revised the introduction to clarify that the study was conducted during the pandemic context to access vulnerable populations instead of examining a direct relationship between COVID-19 and the infections studied. The revised paragraph now reads in lines 76-86:

"The coronavirus disease 2019 (COVID-19) pandemic disproportionally affected economically vulnerable populations, and its peak periods disrupted progress in controlling and preventing STIs. Therefore, WHO recommended taking advantage of COVID-19 testing initiatives to screen vulnerable people for HIV, viral hepatitis, and STIs [8].”

Q1b Why were persons using alcohol or drugs excluded? One could argue that they are a vulnerable group. The population studied seems like a composite group that is "vulnerable" but quite heterogenous Waste collectors, homeless and lgbt are very different worlds and sexual behaviors. There seems to be an opportunistic aspect to this choice, to have a bigger sample size

R1b: As described in lines 90 and 91, we excluded only individuals under psychoactive substances during the data collection because those individuals could not respond appropriately to the research questions. A hundred ninety-nine individuals reported they used illicit drugs, and 29 used alcohol (Table 1).

Concerning the group studied, as described in Methods, during the first waves of the pandemic, our university offered COVID testing free of charge for socially and economically vulnerable individuals assisted by NGO partners. Although we agree they are not homogeneous (considering some risky behaviors), they share health determinants such as low income, low education, and historically low access to health services. In addition, there are overlaps between groups.

Q1c For the models the risk is not necessarily linearly associated with age so I suggest using age categories to compute APR.

R1c: Following your suggestion, we compared the models, including age as a dummy variable (≤ 35 years;> 35 years, based on median age) vs. age as a continuous variable, and the models fit better with age as a continuous variable. Therefore, we ask you to consider the current models.

Q1d In the results the text repeats values and 95%CI that are already in the table. I think it is a bit redundant.

R1d: Thank you for your careful review. We have revised the results section to remove unnecessary repetitions while ensuring the key findings remain clearly described in the text.

Q1e Since we are mostly talking about sexual risk there should be a table with the sexual risks per subpopulation (what proportion of waste collectors used condoms, engaged in anal sex, translational sex...etc for refugees and migrants lgbt....)

Q1e: Thank you for your careful review. We included S1Table, with information about sociodemographic characteristics and sexual risk per group.

Reviewer #2: Dear authors, I have read the article entitled “Screening vulnerable populations for hepatitis B, hepatitis C, syphilis, and human immunodeficiency virus during the coronavirus disease 2019 pandemic”. I have some minor concerns:

Q2a For better understanding, the data description in each table should be given before the tables appear (Lines: 172-178; lines: 212-237).

R2a: Thank you for your careful review. We have revised the manuscript to ensure that the data description precedes the corresponding tables in the specified sections.

Q2b In addition, as each table corresponds to a part of the results, each part must have a title.

R2b: We have revised the manuscript to ensure that each section of the results corresponds to a clearly defined title, aligning with the respective tables.

Q2c Lines 183-186: These data are not shown in Table 2.

R 2c: Thank you for your careful review. HBV DNA and HCV RNA detection results were presented in another paragraph. Please let us know if you agree.

Reviewer #3:

Q3a Line 1 to 3: The authors could specify the study area in the title.

R3a: Thank you for your careful review. We considered your suggestion and explicitly modified the title to include the study area ("Taking the opportunity of COVID testing to screen vulnerable populations for hepatitis B, hepatitis C, syphilis, and human immunodeficiency virus in Goiás, Central Brazil). Please let us know if you agree with this new version.

Q3b Line 23: The 273-word abstract highlights the important results of this study. However, it would be interesting to highlight the context and objectives of the study at the beginning of the summary.

R3b: Thank you for your careful review. We have revised the abstract to highlight the context and objectives of the study at the beginning, ensuring a clearer introduction to the research.

Q3c Harmonize this reference with the previous ones which are in the numbered style.

R3c: Thank you for your careful review. The reference was harmonized as recommended.

3d Line 144 to 147: Which extraction kit was used for HCV RNA extraction?

R3d: We added the kit’s name to the text. QIAamp® Viral RNA Mini Kit; Qiagen, Hilden, Germany.

3e Line 154: Make the P lowercase. Harmonize with the previous one.

R3e: Thank you for your careful review. The p has been harmonized with the other instances throughout the manuscript to ensure consistency.

3f Line 167 (table 1): The sum of the percentages exceeds 100; there are certainly groups that overlap; could you separate them?

R3f: Thank you for your careful review. We included the information on overlap groups in the Table’s Legend.

Q3g Conclusion: The manuscript does not clearly highlight the correlation between COVID-19 and other infections. I therefore suggest that we remove the term from the title.

R3g: Thank you for your careful review. We replaced the title with "Taking the opportunity of COVID testing to screen vulnerable populations for hepatitis B, hepatitis C, syphilis, and human immunodeficiency virus in Central Brazil." This new title ensures that the study is accurately represented while avoiding any implication of a causal relationship between COVID-19 and the infections analyzed.

---

## [Decision Letter · Decision Letter 1]

Taking the opportunity of COVID testing to screen vulnerable populations for hepatitis B, hepatitis C, syphilis, and human immunodeficiency virus in Central Brazil

PONE-D-24-55277R1

Dear Dr. Teles,

We’re pleased to inform you that your manuscript has been judged scientifically suitable for publication and will be formally accepted for publication once it meets all outstanding technical requirements.

Kind regards,

Pierre Roques, Ph.D.

Academic Editor

PLOS ONE

Additional Editor Comments (optional):

The reviewers and the editor thank you to have take in account all suggestions they indicated.

Reviewers' comments:

Reviewer's Responses to Questions

**Comments to the Author**

1. If the authors have adequately addressed your comments raised in a previous round of review and you feel that this manuscript is now acceptable for publication, you may indicate that here to bypass the “Comments to the Author” section, enter your conflict of interest statement in the “Confidential to Editor” section, and submit your "Accept" recommendation.

Reviewer #1: All comments have been addressed

Reviewer #2: All comments have been addressed

2. Is the manuscript technically sound, and do the data support the conclusions?

Reviewer #1: Yes

Reviewer #2: Yes

3. Has the statistical analysis been performed appropriately and rigorously? 

Reviewer #1: Yes

Reviewer #2: I Don't Know

4. Have the authors made all data underlying the findings in their manuscript fully available?

Reviewer #1: Yes

Reviewer #2: Yes

5. Is the manuscript presented in an intelligible fashion and written in standard English?

Reviewer #1: Yes

Reviewer #2: Yes

6. Review Comments to the Author

Reviewer #1: thank you for your efforts to improve the manuscript. i no longer have any thing to add. i hope the manuscript will be cited.

Reviewer #2: (No Response)

7. PLOS authors have the option to publish the peer review history of their article (what does this mean? ). If published, this will include your full peer review and any attached files.

**Do you want your identity to be public for this peer review?** For information about this choice, including consent withdrawal, please see our Privacy Policy .

Reviewer #1: No

Reviewer #2: No

---

## [Editor Report · Acceptance letter]

PONE-D-24-55277R1

PLOS ONE

Dear Dr. Teles,

I'm pleased to inform you that your manuscript has been deemed suitable for publication in PLOS ONE. Congratulations! Your manuscript is now being handed over to our production team.

Kind regards,

on behalf of

Dr. Pierre Roques

Academic Editor

PLOS ONE